# Leveraging the Third Dimension in Contrastive Learning

## Abstract

Self-Supervised Learning (SSL) methods operate on unlabeled data to learn robust representations useful for downstream tasks. Most SSL methods rely on augmentations obtained by transforming the 2D image pixel map. These augmentations ignore the fact that biological vision takes place in an immersive three-dimensional, temporally contiguous environment, and that low-level biological vision relies heavily on depth cues. Using a signal provided by a pretrained state-of-the-art monocular RGB-to-depth model (the *Depth Prediction Transformer*, Ranftl et al., 2021), we explore two distinct approaches to incorporating depth signals into the SSL framework. First, we evaluate contrastive learning using an RGB+depth input representation. Second, we use the depth signal to generate novel views from slightly different camera positions, thereby producing a 3D augmentation for contrastive learning. We evaluate these two approaches on three different SSL methods—BYOL, SimSiam, and SwAV—using ImageNette (10 class subset of ImageNet) and ImageNet-100. We find that both approaches to incorporating depth signals improve the robustness and generalization of the baseline SSL methods, though the first approach (with depth-channel concatenation) is superior. For instance, BYOL with the additional depth channel leads to an increase in downstream classification accuracy from 85.3% to 88.0% on ImageNette and 84.1% to 87.0% on ImageNet-C.

## 1 Introduction

Biological vision systems evolved in and interact with a three-dimensional world. As an individual moves through the environment, the relative distance of objects is indicated by rich signals extracted by the visual system, from motion parallax to binocular disparity to occlusion cues. These signals play a role in early development to bootstrap an infant's ability to perceive objects in visual scenes (Spelke, 1990; Spelke & Kinzler, 2007) and to reason about physical interactions between objects (Baillargeon, 2004). In the mature visual system, features predictive of occlusion and three-dimensional structure are extracted early and in parallel in the visual processing stream (Enns & Rensink, 1990; 1991), and early vision uses monocular cues to rapidly complete partially-occluded objects (Rensink & Enns, 1998) and binocular cues to guide attention (Nakayama & Silverman, 1986). In short, biological vision systems are designed to leverage the three-dimensional structure of the environment.

In contrast, machine vision systems typically consider a 2D RGB image or a sequence of 2D RGB frames to be the relevant signal. Depth is considered as the end product of vision, not a signal that can be exploited to improve visual information processing. Given the bias in favor of end-to-end models, researchers might suppose that if depth were a useful signal, an end-to-end computer vision system would infer depth. Indeed, it's easy to imagine the advantages of depth processing integrated into the visual information processing stream. For example, if foreground objects are segmented from the background scene, neural networks would not make the errors they often do by using short-cut features to classify (e.g., misclassifying a cow at the beach as a whale) (Geirhos et al., 2020).

In this work, we take seriously the insight from biological vision that depth signals are extracted early in the processing stream, and we explore how depth signals might support computer vision. We assume the availability of a depth signal by using an existing state-of-the-art monocular RGB-to-depth extraction model, the *Dense Prediction Transformer (DPT)* (Ranftl et al., 2021).

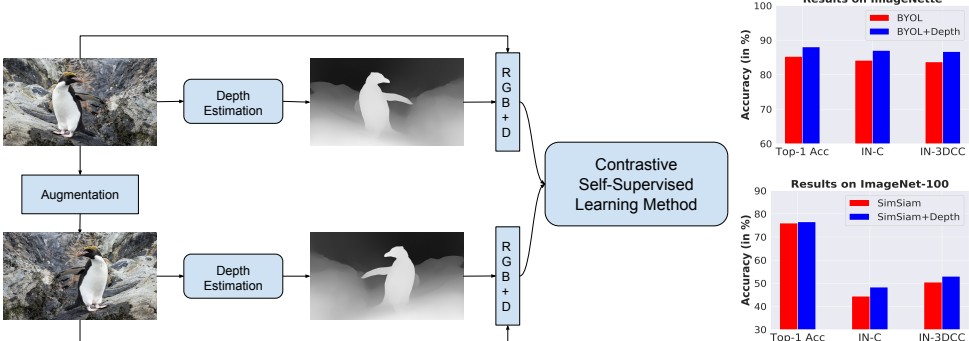

Figure 1: Improving Self-Supervised Learning by concatenating an input channel with estimated depth to the RGB input. Depth is estimated from both an original image and an augmentation, and the resulting 4-channel inputs are used to produce the representation. Incorporating the depth channel improves downstream accuracy in a variety of SSL techniques, with the largest improvements on challenging corrupted benchmarks. (Teaser results are shown. Complete results in Tables 1, 2)

We focus on using the additional depth information for self-supervised representation learning. SSL aims to learn effective representations from unlabelled data that will be useful for downstream tasks (Chen et al., 2020a). We investigate two specific hypotheses. First, we consider directly appending the depth channel to the RGB and then use the RGB+D input directly in contrastive learning (Fig. 1). Second, we consider synthesizing novel image views from the RGB+D representation using a recent method, AdaMPI (Han et al., 2022) and treating these synthetic views as image augmentations for contrastive learning (Fig. 2).

Prior work has explored the benefit of depth signals in supervised learning for specific tasks like object detection and semantic segmentation (Cao et al., 2016; Hoyer et al., 2021; Song et al., 2021; Seichter et al., 2021). Here, we pursue a similar approach in contrastive learning, where the goal is to learn robust, universal representations that support downstream tasks. To the best of our knowledge, only one previous paper has explored the use of depth for contrastive learning (Tian et al., 2020). In their case, ground truth depth was used and it was considered as one of many distinct "views" of the world. We summarize our contributions below:

- Motivated by biological vision systems, we propose two distinct approaches to improving SSL using a (noisy) depth signal extracted from a monocular RGB image. First, we concatenate the derived depth map and the image and pass the four-channel RGB+D input to the SSL method. Second, we use a single-view view synthesis method that utilizes the depth map as input to generate novel 3D views and provides them as augmentations for contrastive learning.

- We show that both of these approaches improve the performance of three different contrastive learning methods (BYOL, SimSiam, and SwAV) on both ImageNette and ImageNet-100 datasets. Our approaches can be integrated into any contrastive learning framework without incurring any significant computational cost and trained with the same hyperparameters as the base contrastive method. We achieve a 2.8% gain in the performance of BYOL with the addition of depth channel on ImageNette dataset.

- Both approaches also yield representations that are more robust to image corruptions than the baseline SSL methods, as reflected in performance on ImageNet-C and ImageNet-3DCC. On the large-scale ImageNet-100 dataset, SimSiam+Depth outperforms base SimSiam model by 4% in terms of corruption robustness.

## 2 RELATED WORK

**Self-Supervised Learning.** The goal of self-supervised learning based methods is to learn a universal representation that can generalize to various downstream tasks. Earlier work on SSL relied on handcrafted pretext tasks like rotation (Gidaris et al., 2018), colorization (Zhang et al., 2016) and jigsaw (Noroozi & Favaro, 2016). Recently, most of the state-of-the-art methods in SSL are based on contrastive representation learning. The goal of contrastive representation learning is to make the representations between two augmented views of the scene similar and also to make representations of views of different scenes dissimilar.

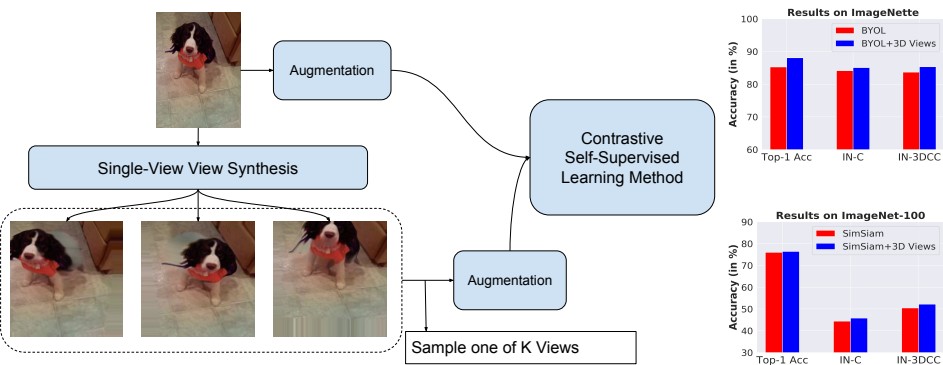

Figure 2: Novel views can be synthesized from a single image by using the estimated depth channel, which can be used as additional augmentations across a variety of contrastive self-supervised learning techniques. These improve results, especially on benchmarks with image corruptions. (Result highlights are shown. Complete results in Tables 1, 2

*SimCLR* (Chen et al., 2020b) showed that augmentations play a key role in contrastive learning and the set of augmentations proposed in the work showed that contrastive learning can perform really well on large-scale datasets like ImageNet. *BYOL* (Grill et al., 2020) is one of the first contrastive learning based methods without negative pairs. BYOL is trained with two networks that have the same architecture: an online network and a target network. From an image, two augmented views are generated; one is routed to the online network, the other to the target network. The model learns by predicting the output of the one view from the other view. *SwAV* (Caron et al., 2020) is an online clustering based method that compares cluster assignments from multiple views. The cluster assignments (or code) from one augmented view of the image is predicted from the other augmented view. *SimSiam* (Chen & He, 2021) explores the role of Siamese networks in contrastive learning. SimSiam is an conceptually simple method as it does not require a BYOL-like momentum encoder or a SwAV-like clustering mechanism.

Contrastive Multiview Coding (CMC) Tian et al. (2020) proposes a framework for multiview contrastive learning that maximizes the mutual information between views of the same scenes. Each view can be an additional sensory signal like depth, optical flow, or surface normals. CMC is closely related to our work but differs in two primary ways. First, CMC considers depth as a separate view and applies a mutual information maximization loss across multiple views; in contrast, we either concatenate the estimated depth information to the RGB input or generate 3D realistic views using the depth signal. Second, CMC considers only ground truth depth maps whereas we show that depth maps estimated from RGB are also quite helpful.

**Monocular Depth Estimation in Computer Vision**. Monocular depth estimation is a pixel-level task that aims to predict the distance of every pixel from the camera using a single image. Though monocular depth estimation is a highly ill-posed problem, deep learning based techniques have been shown to perform extremely well on this task. A few works (Eitel et al., 2015; Cao et al., 2016; Hoyer et al., 2021; Song et al., 2021; Seichter et al., 2021) have explored the benefits of depth estimation for semantic segmentation and object detection. Cao et al. (2016) were one of the first efforts to perform a detailed analysis showing that augmenting the RGB input with estimated depth map can significantly improve the performance on object detection and segmentation tasks. A multi-task training procedure of predicting the depth signal along with the semantic label was also proposed in Cao et al. (2016). RGB-D segmentation with ground truth depth maps was shown to be superior compared to standard RGB segmentation (Seichter et al., 2021). Hoyer et al. (2021) proposed to use self-supervised depth estimation as an auxiliary task for semantic segmentation. Multimodal Estimated-Depth Unification with Self-Attention (MEDUSA) Song et al. (2021) incorporated inferred depth maps with RGB images in a multimodal transformer for object detection tasks. With limited analysis on CIFAR-10, He (2017) showed that estimated depth maps aid image classification.

Most prior works that utilize depth information do so with the objective of improving certain tasks like object detection or semantic segmentation. To the best of our knowledge, ours is the first work that focuses specifically on using an estimated depth signal to enhance contrastive learning. The deep

encoder obtained from contrastive learning can then be used for various downstream tasks like object detection or image classification.

# 3 DEPTH IN CONTRASTIVE LEARNING

We propose two general methods of incorporating depth information into any SSL framework. Both of these methods, which we describe in detail shortly, assume the availability of a depth signal. We obtain this signal from an off-the-shelf pretrained Monocular Depth Estimation model. We generate depth maps for every RGB image in our data set using the state-of-the-art Dense Prediction Transformer (DPT) Ranftl et al. (2021) trained for the monocular depth estimation task. DPT is trained on a large training dataset with 1.4 million images and leverages the power of Vision Transformers. DPT outperforms other monocular depth estimation methods by a significant margin. It has been shown that DPT can accurately predict depth maps for in-the-wild images Han et al. (2022). We treat the availability of these depth maps for contrastive learning as being similar to the availability that people have to extract depth cues via binocular disparity, motion parallax, or occlusion.

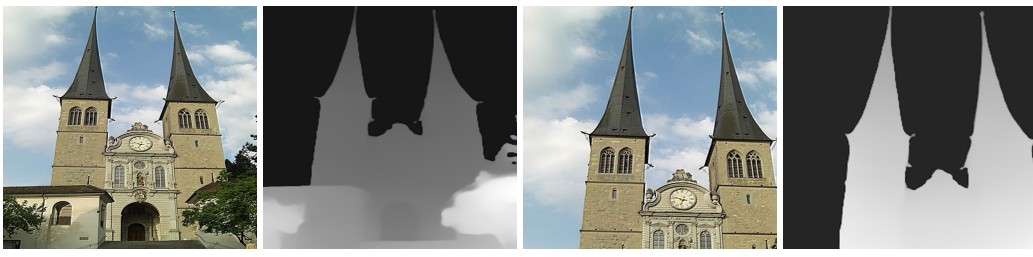

(a) Original Image     (b) Estimated Depth Map     (c) Cropped Image     (d) Estimated Depth Map

Figure 3: Despite two images of a church (Imagenette) being quite similar visually, the presence of a tree occluding the church is a strong hint that the church is in the background, resulting in a very different depth map.

## 3.1 CONCATENATING A DEPTH CHANNEL TO THE INPUT

We analyze the effect of concatenating a depth channel to the RGB image as a means of providing a richer input. This four-channel input is then fed through the model backbone. As we argued earlier, ample evidence suggests that cues to the three dimensional structure of the world are critical in the course of human development (e.g., learning about objects and their relationships), and these cues are available to biological systems early in the visual processing stream and are very likely used to segment the world into objects. Consequently, we hypothesize that a depth channel will support improved representations in contrastive learning.

We anticipate that the depth channel might particularly assist the model when an image is corrupted, occluded, or viewed from an unusual perspective (Fig. 3). Depth might also be helpful in low-light environments where surface features of an object may not be clearly visible. This is quite important in safety critical applications like autonomous driving. The conjecture that depth cues will support interpretation of corrupted images is far from obvious because when the depth estimation method is applied to a corrupted image, the resulting depth maps are less than accurate (see Fig. 6 and 7). We conduct evaluations using two corruption-robustness benchmarks to determine whether the depth signal extracted yields representations that on balance improve accuracy in a downstream classification task. Sample visualizations of the images and their depth map can be found in App. C.

As Figure 1 depicts, our proposed method processes each image and each augmentation of an image through the DPT depth extractor. However, in accord with practice in SSL, we sample a new augmentation on each training step and the computational cost of running DPT on every augmentation in every batch is high. To avoid this high cost of training, we perform a one-time computation of depth maps for every image in the dataset and use this cached map in training for the original image, but we also transform it for the augmentation. This transformation works as follows. First, an augmentation is chosen from the set of augmentations defined by the base SSL method, and the RGB image is transformed according to this augmentation. For the depth map, only the corresponding Random

Crop and Horizontal Flip transforms (i.e., dilation, translation, and rotations) are applied. The resulting depth map for the augmentation is cheap to compute, but it has a stronger correspondence to the original image's depth map than one might expect had the depth map been computed for the augmentation by DPT. To address the possibility that the SSL method might come to rely too heavily on the depth map, we incorporated the notion of *depth dropout*.

With depth dropout, the depth channel of any original image or augmentation is cleared (set to 0) with probability $p$, independently decided for each image or augmentation. When depth dropout is integrated with a SSL method, it prevents the SSL method from becoming too dependent on the depth signal by reducing the reliability of that signal. Consider a method like BYOL, whose objective is to predict the representations of one view from the other. With depth dropout, the objective is much more challenging. Since the depth channel is dropped out in some views, the network has to learn to predict the representations of a view with a depth signal using a view without depth. This leads to the model capturing additional 3D structure about the input without any significant computation cost.

At evaluation, every image in the evaluation set *is* processed by DPT; the short cut of remapping the depth channel from the original image to the augmentation was used only during training.

## 3.2 3D VIEWS WITH ADAMPI

We now discuss our second method of incorporating depth information in contrastive SSL methods. This method is motivated by the fact that humans have two eyes and binocular vision requires us to match up the different views of the world seen by each eye. Because each eye has a subtlely different perspective, the images impinging on the retina are slightly different. The brain integrates the two images by determining the correspondence between regions from each eye. This *stereo correspondence* helps people in understanding and representing the 3D scene. We introduce this idea into Self-Supervised Learning with the help of Single-View View Synthesis methods.

Single-View View Synthesis (Tucker & Snavely, 2020) is an extreme version of the view synthesis problem that takes single image as the input and renders images of the scene from new viewpoints. The task of view synthesis requires a deep understanding of the objects, scene geometry and appearance. Most of the methods proposed for this task make use of multiplane-image (MPI) representation (Tucker & Snavely, 2020; Li et al., 2021; Han et al., 2022). MPI consists of $N$ fronto-parallel RGB$\alpha$ planes arranged at increasing depths. MINE (Li et al., 2021) introduced the idea of Neural Radiance Fields (Mildenhall et al., 2020) into the MPI to perform novel view synthesis with a single image. These single-view view synthesis methods have a wide ranging applications in Augmented and Virtual Reality as they allow the viewer to interact with the photos.

Recently, a lot of single-view view synthesis methods have been using layered depth representations (Shih et al., 2020; Jampani et al., 2021). These methods have been shown to generalize well on the unseen real world images. As mentioned in Section 3.1, monocular depth estimation models like DPT (Ranftl et al., 2021) are used when depth maps are not available. AdaMPI (Han et al., 2022) is one such recently proposed method that aims to generate novel views for in-the-wild images. AdaMPI introduces two novel modules, a plane adjustment network and a color prediction network to adapt to diverse scenes. Results show that AdaMPI outperforms MINE and other single image view synthesis methods in terms of quality of the synthesized images. We use AdaMPI for all of the experiments in our paper, given the quality of synthesized images generated by AdaMPI.

At inference, AdaMPI takes an RGB image, depth (estimated from the monocular depth estimation model), and the target view to be rendered. The single-view view synthesis model then generates a multiplane-image representation of the scene. This representation can then be easily used to transform the image in the source view to the target view. More details about AdaMPI is present in App. B.

In a nutshell, AdaMPI generates a "3D photo" of a given scene given a single input. In a way, it can be claimed that an image can be "brought to life" by generating the same image from another camera viewpoint (Kopf et al., 2019). We propose to use the views generated by AdaMPI as augmentations for SSL methods (Fig. 2). The synthesized views captures the 3D scene and generates realistic augmentations that help the model learn better representations. These augmentations are meant to reflect the type of subtle shifts in perspective obtained from the two eyes or from minor head or body movements.

Table 1: Results on ImageNette Dataset show consistently improved robustness from explicitly leveraging depth estimation. Additionally, the depth channel approach consistently outperforms the 3D view augmentation approach.

| Method | $k$NN | Top-1 Acc. | ImageNet-C | ImageNet-3DCC |
|---|---|---|---|---|
| BYOL (Grill et al., 2020) | 85.71 | 85.27 | 84.13 | 83.68 |
| + Depth ($p = 0.5$) | 88.56 | 88.03 | 87.00 | 86.68 |
| + 3D Views | 87.01 | 87.42 | 85.75 | 85.86 |
| SimSiam (Chen & He, 2021) | 85.10 | 85.76 | 84.08 | 84.16 |
| + Depth ($p = 0.5$) | 86.52 | 87.41 | 85.13 | 85.08 |
| + 3D Views | 85.94 | 87.62 | 83.87 | 84.37 |
| SwAV (Caron et al., 2020) | 89.63 | 91.08 | 75.31 | 82.05 |
| + Depth ($p = 0.5$) | 89.20 | 90.85 | 83.80 | 85.02 |

Augmentations are a key ingredient in contrastive learning methods (Chen et al., 2020a). Modifying the strength of augmentations or removing certain augmentations leads to significant drop in the performance of contrastive methods (Chen et al., 2020a; Grill et al., 2020; Zhang & Ma, 2022). Most of these augmentations can be considered as "2D" as they make changes in the image either by cropping the image or applying color jitter. On the other hand, the generated 3D views are quite diverse as they bring in another dimension to the contrastive setup. Moreover, they can be combined with the existing set of augmentations to achieve the best performance.

The synthesized views as augmentations allow the model to virtually interact with the 3D world. For every training sample, we generate $k$ views synthesized from the camera in the range of $x$-axis range, $y$-axis range and $z$-axis range. The $x$-axis range essentially refers to the shift in the $x$-axis from the position of the original camera. The synthesis of the 3D Views is computed only once for the training dataset in an offline manner. Out of the total $k$ views per sample, we sample one view at every training step and use it for training. We tried two techniques to augment the synthesized views. First, we applied the augmentations of the base SSL method on top of the synthesized view. Second, we applied the base SSL augmentations with a probability of $q$ or we used the synthesized view (with Random Crop and Flip) with a probability of 1-$q$. Full details can be found in the Appendix.

The range of novel camera views generated by the single-view view synthesis method can be controlled by the user. It is possible to specifically control the $x$-axis shift, $y$-axis shift and $z$-axis shift (zoom) during the generation of the novel views. The quality of generated images degrades when the novel view to be generated is far from the current position of the camera. This is expected because it is not feasible to generate a complete 360-degree view of the scene by using a single image. In practice, we observe certain artifacts in the image when views far away from the current position of the camera. Additional details can be found in App. A and App. D.

## 4 EXPERIMENTAL RESULTS

We show results with the addition of depth channel and 3D Views with various SSL methods on ImageNette and ImageNet-100 datasets. We also measure the corruption robustness of these models by evaluating the performance of these models on ImageNet-C and ImageNet-3DCC.

### 4.1 EXPERIMENTAL SETUP

**ImageNette**: is a 10 class subset of ImageNet (Deng et al., 2009) that consists of 9469 images for training and 2425 images for testing. We use the 160px version of the dataset for all the experiments and train the models with an image size of 128.

**ImageNet-100**: is a 100 class subset of ImageNet (Deng et al., 2009) consisting of 126689 training images and 5000 validation images. We use the same classes as in (Tian et al., 2020) and train all models with image size of 224.

**ImageNet-C** (IN-C) (Hendrycks & Dietterich, 2019): ImageNet-C dataset is a benchmark to evaluate the robustness of the model to common corruptions. It consists of 15 types of algorithmically

Table 2: Results on ImageNet-100 Dataset indicates that both addition of the depth channel and 3D Views leads to a gain in corruption robustness performance.

| Method | $k$NN | Top-1 Acc. | ImageNet-C | ImageNet-3DCC |
|---|---|---|---|---|
| BYOL (Grill et al., 2020) | 74.24 | 80.74 | 47.15 | 53.69 |
| + Depth (p = 0.3) | 74.66 | 80.24 | 50.17 | 55.55 |
| + 3D Views | 73.42 | 80.16 | 48.15 | 54.88 |
| SimSiam (Chen & He, 2021) | 67.56 | 76.00 | 44.39 | 50.44 |
| + Depth (p = 0.2) | 70.90 | 76.54 | 48.30 | 52.93 |
| + 3D Views | 68.08 | 76.40 | 45.78 | 52.17 |

generated corruptions including weather corruptions, noise corruptions and blur corruptions with different severity. Refer to Fig. 6 for a visual depiction of the images corrupted with Gaussian Noise.

**ImageNet-3DCC** (IN-3DCC) (Kar et al., 2022): ImageNet-3DCC consists of realistic 3D corruptions like camera motion, occlusions, weather to name a few. The 3D realistic corruptions are generated using the estimated depth map and improves upon the corruptions in ImageNet-C. Some examples of these corruptions include XY-Motion Blur, Near Focus, Flash, Fog3D to name a few.

**Experimental Details.** We use a ResNet-18 (He et al., 2016) backbone for all our experiments. For the pretraining stage, the network is trained using the SGD optimizer with a momentum of 0.9 and batch size of 256. The ImageNette experiments are trained with a learning rate of 0.06 for 800 epochs whereas the ImageNet-100 experiments are trained with a learning rate of 0.2 for 200 epochs. We implement our methods in PyTorch 1.11 (Paszke et al., 2019) and use Weights and Biases (Biewald, 2020) to track the experiments. We refer to the *lightly* (Susmelj et al., 2020) benchmark for ImageNette experiments and *solo-learn* (da Costa et al., 2022) benchmark for ImageNet-100 experiments. We follow the commonly used linear evaluation protocol to evaluate the representations learned by the SSL method. For linear evaluation, we use SGD optimizer with a momentum of 0.9 and train the network for 100 epochs. For the ImageNette+3D Views experiments, we apply base SSL augmentation on top of the synthesized views at every training step. For the ImageNet-100+3D Views experiments, we apply the base SSL augmentations with a probability of 0.5. Additional experimental details is present in the App. A.

## 4.2 Results on ImageNette

Table 1 shows the benefit of incorporating depth with any SSL method on the ImageNette dataset. We use the $k$-nearest neighbor ($k$NN) classifier and Top-1 Acc from the linear evaluation performance to evaluate the learned representation of the SSL method. It can be seen that the addition of depth improves the accuracy of BYOL, SimSiam and SwAV. BYOL+Depth indicates that the model is trained with depth map with the depth dropout. BYOL+Depth improves upon the Top-1 accuracy of BYOL by 2.8% along with a 3% increase in the ImageNet-C and ImageNet-3DCC performance. This clearly demonstrates the role of depth information in corrupted images.

We observe a significant 8.5% increase in the ImageNet-C with SwAV+Depth over the base SwAV. On a closer look, it can be seen that the addition of depth channel results in high robustness to noise-based perturbations and blur-based perturbations. For instance, the accuracy on the Motion Blur corruption increases from 70.32% with SwAV to 86.88% with SwAV+Depth. And the performance on Gaussian Noise corruption increases from 69.76% to 84.56% with the addition of depth channel.

BYOL + 3D Views indicates that the views synthesized by AdaMPI are used as augmentations in the contrastive learning setup. We show that proposed 3D Views leads to a gain in accuracy with both BYOL and SimSiam. This indicates that the diversity in the augmentations due to the 3D Views helps the model capture a better representation of the world. We also observe a decent gain in accuracy on IN-C and IN-3DCC with 3D views compared to the baseline BYOL.

## 4.3 Results on ImageNet-100

Table 2 summarizes the results on the large-scale ImageNet-100 with BYOL and SimSiam. We find that most of the observations on the ImageNette datasets also hold true in the ImageNet-100 datasets. Though the increase in the Top-1 Accuracy with the inclusion of depth is minimal, we observe that

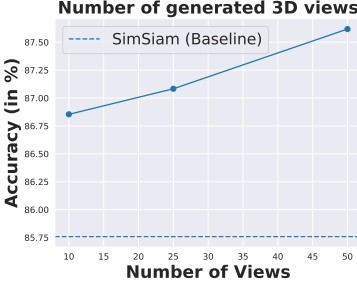

**Figure 4:** As the number of 3D views increases, the performance of the SSL method increases with very limited increase in performance.

| Method | Top-1 Acc. | IN-C | IN-3DCC |
|---|---|---|---|
| BYOL (Grill et al., 2020) | 85.27 | 84.13 | 83.68 |
| + Depth ($p = 0.0$) | 84.38 | 72.64 | 73.68 |
| + Depth ($p = 0.2$) | 89.05 | 85.93 | 85.33 |
| + Depth ($p = 0.5$) | 88.03 | 87.00 | 86.68 |
| + Depth ($p = 0.8$) | 86.57 | 85.38 | 85.60 |

Table 3: Ablation of Depth Dropout hyperparameter ($p$). A large dropout ($p = 0.8$) leads to the model ignoring the depth signal and a low (or zero) depth dropout leads to model relying only on depth signal.

performance on ImageNet-C and ImageNet-3DCC increases notably. With SimSiam, we notice a 3.9% increase in ImageNet-C accuracy and a 2.5% increase in ImageNet-3DCC accuracy just by the addition of depth channel. These results emphasize the role of the proposed depth channel with dropout in contrastive learning.

We observe that the proposed method of incorporating 3D views outperforms the base SSL method on the ImageNet-100 dataset, primarily in the corruption benchmarks. On a detailed look at the performance of each corruption, we observe that the 3D Views improves the performance of 3D based corruptions by more than 2.5%. (Refer Table 4)

## 5 DISCUSSION

**Depth Dropout**. Table 3 shows the ablation of probability of Depth dropout ($p$) on the ImageNette dataset with BYOL. The influence of using the depth dropout can also be understood with these results. It can be observed that without depth dropout ($p = 0.0$), the performance of the model is significantly lower than the baseline BYOL, as the network learns to focus solely on the depth channel. We find that $p = 0.2$ leads to the highest Top-1 Accuracy but $p = 0.5$ achieves the best performance on the ImageNet-C and ImageNet-3DCC. As the depth dropout increases (p = 0.8), the performance gets closer to the base SSL method as the model completely ignores the depth channel.

**What happens when depth is not available during inference?** In this ablation, we examine the importance of depth signal at inference. Given a model trained with depth information, we analyze what happens when we set the depth to 0 at inference. Table 6 reports these results on ImageNette dataset with BYOL. Interestingly, we find that even with the absence of depth information, the accuracy of the model is higher than the baseline BYOL. This indicates that the model has implicitly learned some depth signal and captured better representations. It can also be seen that the performance on IN-3DCC is 1.5% higher than BYOL. Furthermore, we observe that the addition of depth map improves the performance on all the benchmarks. This further highlights our message that depth signal is a useful signal in learning a robust model.

**Number of Views generated by AdaMPI**. Figure 4 investigates the impact of the number of generated 3D views on the performance of SimSiam (ImageNette). We observe that as the number of views increases, the Top-1 Accuracy increases although the gains are quite minimal. It must be noted that even with 10 views, the SimSiam+3D Views outperforms the baseline SimSiam by 1.5%.

**Which corruptions improve due to depth and 3D Views?** A detailed analysis of the performance of the methods on various type of corruptions is reported in Table 4. We report the average on different categories of corruptions to understand the role of various corruptions on the overall performance. For ImageNet-C (IN-C), we divide the corruptions into 4 groups: Noise, Blur, Weather and Digital. ImageNet-3DCC is split up into two categories based on whether they make use of 3D information. We observe that the depth channel leads to a massive 5.7% average gain on the noise corruptions and 3.4% increase in digital corruptions over the baseline. The use of 3D Views in SSL results in a notable 4.2% improvement on the Blur corruptions over the base SSL method. As expected, the performance on 3D Corruptions with the 3D Views is much higher than standard SSL method and slightly higher than the method that uses depth channel. More results can be found in App. E.

Table 4: Results on ImageNet-100 Corruptions show that while use of 3D view augmentations provides a larger improvement on 3D corruptions, the improvements from using depth channel are more consistent on a wide range of corruptions. Detailed results in App. E.

| Method | IN-C | Noise | Blur | Weather | Digital | IN-3DCC | 3D | Misc |
|---|---|---|---|---|---|---|---|---|
| BYOL (Grill et al., 2020) | 47.15 | 36.69 | 38.95 | 49.57 | 59.33 | 53.69 | 54.53 | 51.16 |
| + Depth (p = 0.3) | 50.17 | 42.36 | 40.66 | 51.88 | 62.17 | 55.55 | 55.85 | 54.65 |
| + 3D Views | 48.15 | 34.50 | 43.06 | 50.16 | 60.14 | 54.88 | 56.56 | 49.81 |
| SimSiam (Chen & He, 2021) | 44.39 | 36.20 | 36.11 | 45.24 | 55.86 | 50.44 | 51.32 | 47.83 |
| + Depth (p = 0.2) | 48.30 | 41.90 | 38.40 | 49.76 | 59.84 | 52.93 | 53.16 | 52.25 |
| + 3D Views | 45.78 | 35.00 | 40.42 | 46.20 | 57.14 | 52.17 | 53.69 | 47.63 |

Table 6: These results on ImageNette show that the model is robust to the absence of depth signal and that estimated depth improves the corruption robustness and linear evaluation performance.

| Method | Top-1 Acc. | IN-C | IN-3DCC |
|---|---|---|---|
| BYOL (Grill et al., 2020) | 85.27 | 84.13 | 83.68 |
| + Depth ($p = 0.5$) | 88.03 | 87.00 | 86.68 |
| Depth = 0 at inference | 86.80 | 84.95 | 85.21 |

Table 7: Comparison of two Single-View View Synthesis Methods for generating 3D Views on ImageNette dataset. Higher quality views leads to higher performance.

| Method | Top-1 Acc. | IN-C | IN-3DCC |
|---|---|---|---|
| BYOL (Grill et al., 2020) | 85.27 | 84.13 | 83.68 |
| + 3D Views (MINE) | 87.49 | 84.47 | 83.93 |
| + 3D Views (AdaMPI) | 88.08 | 85.07 | 85.33 |

**Range of Views generated by AdaMPI**. The range of 3D Views generated by AdaMPI play a huge role in the performance of the SSL method. Table 5 summarizes the effects of moving the target camera on the learned representations on ImageNette dataset. $x$ denotes the amount by which the $x$-axis is moved and $y$ denotes the same for $y$-axis. We observe that a very small change in viewing

Table 5: Ablation on Range of synthesized views generated by AdaMPI. Results are shown on ImageNette dataset.

| Method | Top-1 Acc. | IN-C | IN-3DCC |
|---|---|---|---|
| BYOL (Grill et al., 2020) | 85.27 | 84.13 | 83.68 |
| + 3D Views ($x = 0.1; y = 0.1$) | 86.09 | 83.33 | 83.63 |
| + 3D Views ($x = 0.4; y = 0.4$) | 87.87 | 84.78 | 85.22 |
| + 3D Views ($x = 0.5; y = 0.5$) | 88.08 | 85.07 | 85.33 |
| + 3D Views ($x = 0.8; y = 0.8$) | 87.49 | 82.47 | 84.35 |
| + 3D Views ($x = 1.0; y = 1.0$) | 86.34 | 80.81 | 83.30 |

direction ($x = 0.1; y = 0.1$) does not boost the performance very much. As $x$ and $y$ get larger, the quality of generated images also decreases. Thus, a large change in the viewing direction leads to artifacts which hurts the performance. This can be clearly observed in Table 5 where we see a drop in accuracy as the $x$ and $y$ increases from 0.5 to 1.0.

**Quality of Synthesized Views**. In this ablation, we investigate how the quality of the synthesized views affects the representations learnt by Self-Supervised methods. We compare two different methods to generate 3D Views of the image namely MINE (Li et al., 2021) and AdaMPI (Han et al., 2022). The quantitative and qualitative results shown in Han et al. (2022) indicate that AdaMPI generates superior quality images compared to MINE. Table 7 reports the results on ImageNette with BYOL comparing the 3D Views synthesized by MINE and AdaMPI methods. We observe that the method with 3D Views generated by AdaMPI outperforms the method with 3D Views generated by MINE. This is a clear indication that *as the quality of 3D view synthesis methods improves, the accuracy of the SSL methods with 3D views increases as well*.

## 6 CONCLUSION

In this work, we propose two distinct approaches to improving SSL using a (noisy) depth signal extracted from a monocular RGB image. Our results on ImageNette and ImageNet-100 datasets with a range of SSL methods (BYOL, SimSiam and SwAV) show that both proposed approaches outperform the baseline SSL on test accuracy and corruption robustness. Further, our approaches can be integrated into any SSL method to boost performance. We close with several critical directions for future research. First, given that our two approaches are complementary and compatible, we might evaluate the two approaches in combination. Second, is depth dropout necessary when depth extraction with DPT can be run on every augmentation on every training step? Third, one might explore the idea of synthesizing views in Single-View View Synthesis methods with the goal of maximizing the performance (Ge et al., 2022) or develop better methods to utilize the 3D Views.

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

## A    EXPERIMENTAL DETAILS

We discuss the detailed experimental setup to allow reproducibility of the results.

**Pretraining**:

**BYOL**: The architecture of the online and target networks in BYOL consists of three components: encoder, projector and predictor. We use ResNet-18 (He et al., 2016) implementation available in *torchvision* as our encoder. The Prediction Network in BYOL is a Multi-Layer Perceptron (MLP) that consists of a linear layer with an output dimension of 4096, followed by Batch Normalization (Ioffe & Szegedy, 2015), ReLU (Nair & Hinton, 2010) and a final linear layer with a dimension of 256. We use the same augmentations as in *lightly* benchmark which uses a slightly modified version of augmentations used in SimCLR (Chen et al., 2020b). The network is trained with an SGD Optimizer with a momentum of 0.9 and a weight decay of 0.0005. A batch size of 256 is used and the network is trained for a total of 800 epochs with a cosine annealing scheduler.

For ImageNet-100, we use the ResNet-18 encoder and train the network using an SGD optimizer with a momentum of 0.9 and a weight decay of 0.0001. We use the set of augmentations in *solo-learn* benchmark in our experiments. The model is trained for 200 epochs with a batch size of 256. The architecture of the prediction head is same as the one used in ImageNette but with the output dimension of the linear layer set to 8192.

**SimSiam** We follow the same optimization hyperparameters as in BYOL for the ImageNette dataset. The architecture of the projection head is a 3-layer MLP with Batch Normalization and ReLU applied to each layer. (The output layer does not have ReLU). The prediction head is a 2-layer MLP with a hidden dimension of 512.

**SwAV**: For SwAV, we use the Adam optimizer (Kingma & Ba, 2014) with a learning rate of 0.001 and weight decay of 0.000001. The number of code vectors (or prototypes) is set to 3K with 128 dimensions. The projection head is a 2-layer MLP with a hidden layer dimension of 2048 and an output dimension of 128. SwAV also introduced the idea of multi-crop where a single input image is transformed into 2 global views and $V$ local views. 6 local views are used in our ImageNette experiments.

**Linear Probing**:

For linear probing, we choose the model with the highest validation $k$NN accuracy and freeze the representations. We then train a linear layer using SGD with momentum optimizer for 100 epochs. We do a grid search on $\{0.2, 0.5, 0.8, 5.0\}$ and report the best accuracy of the best performing model. This is commonly followed in the SSL literature (Zhou et al., 2021). We use the standard set of augmentations which includes Random Resized Crop and Horizontal Flip for training. For ImageNet-100, we observe that a higher learning rate seems to help and we do a grid search on $\{0.5, 0.8, 5.0, 30.0\}$. In most of the experiments, we observe that using the learning rate of 30.0 yields the best-performing model.

**Depth Prediction Transformer**

We refer to the official implementation of the DPT [1] to compute the depth maps. The weights of the best-performing monocular depth estimation model i.e, *DPT-Large*, is used for the calculation of the depth maps. We use the relative depth maps generated by the DPT model.

**AdaMPI**:

We refer to the official implementation of AdaMPI [2] paper to compute the 3D Views. The depth maps generated by DPT are fed as input to the AdaMPI. We generate 50 views per sample. A pretrained AdaMPI model with 64 MPI planes is used in our experiments.

For the ImageNette experiments, we apply base SSL augmentations on top of the generated AdaMPI at every training step. We did a grid search on a set of generated views and selected the best performing model. For both BYOL and SimSiam $x = 0.4$; $y = 0.4$ and $z = 0.0$ was used to generate 3D Views.

---

[1] https://github.com/isl-org/DPT
[2] https://github.com/yxuhan/AdaMPI

For ImageNet-100, we apply the base SSL augmentations with a probability of 0.5 and use the synthesized views with a probability of 0.5. We use the views synthesized with $x = 0.2$; $y = 0.2$ and $z = 0.2$.

For ImageNet-100 experiments, we use Automatic Mixed Precision training to speed up the training. All the ImageNette experiments are run on RTX 8000 GPUs while the ImageNet-100 experiments are run on A100 GPUs. We are thankful to the authors of DPT (Ranftl et al., 2021) and AdaMPI (Han et al., 2022) for publicly releasing the code and pretrained weights. We will also release the code and pretrained weights to enable reproducible research.

## B  ADAMPI

This section explains about how AdaMPI renders new views. The notation and content of this section is heavily derived from Han et al. (2022) and Li et al. (2021).

Consider a pixel coordinate in a image as $[x, y]$, the camera intrinsic matrix $K$, camera rotation matrix $R$, camera translation matrix $t$. A Multiplane image (MPI) is a layered representation that consists of $N$ fronto-parallel RGB$\alpha$ planes arranged in the increasing order of depth.

The first step in rendering a novel view to find the correspondence between the source pixel coordinates $[x_s, y_s]^T$ and target pixel coordinates $[x_t, y_t]^T$. This can be done by using the homography function (Hartley & Zisserman, 2004) as shown by the equation below.

$$\left[x_s, y_s, 1\right]^\top \sim \mathrm{K}\left(\mathrm{R} - \frac{\mathbf{tn}^\top}{d_i}\right)\mathrm{K}^{-1}\left[x_t, y_t, 1\right]^\top, \tag{1}$$

where, $\mathbf{n} = [0, 0, 1]^\top$ is the normal vector of the fronto-parallel plane in the source view. Equation 1 essentially maps the correspondence between source and target pixel coordinate at a particular MPI plane.

The plane projections at the target plane $c'_{d_i}(x_t, y_t) = c'_{d_i}(x_s, y_s)$ and $\sigma'_{d_i}(x_t, y_t) = \sigma'_{d_i}(x_s, y_s)$. Volume rendering (Li et al., 2021; Kajiya & Von Herzen, 1984; Mildenhall et al., 2020) and Alpha compositing can then be used to render the image.

AdaMPI has two major components, a planar adjustment network and color prediction network. In previous works Tucker & Snavely (2020), the $d_i$ was usually fixed. However, in AdaMPI, the planar adjustment predicts $d_i$ and each MPI plane at correct depth. The color prediction network takes this adjusted depth planes and predicts the color and density at each plane. For additional details, we refer the reader to Han et al. (2022).

## C  VISUALIZATION OF DEPTH MAPS

In this section, we show sample visualization of the depth map generated by the DPT model. Figure 5 shows some sample visualization of the original image and the corresponding depth maps. The impact of corrupted images on the estimated depth maps is shown in Fig. 7. It can be seen that high severity in Gaussian Noise distorts the estimated depth maps significantly.

In Figure 3, we show the impact of occlusion on the estimated depth map. Fig 3a contains a tree in front of it and thus it looks like the Church building has a low depth (It is far away). When we just crop the image and remove the trees (Fig. 3c), it can clearly seen how the estimated depth maps changes drastically (Fig. 3d).

## D  VISUALIZATION OF 3D VIEWS

We refer the reader to the supplementary zip file for some sample videos and images of synthesized views from AdaMPI.

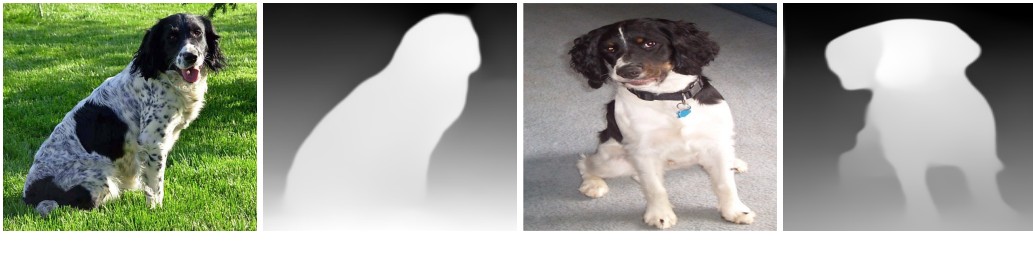

(a) Original Image     (b) Estimated Depth Map     (c) Original Image     (d) Estimated Depth Map

Figure 5:  Visualization of Depth Maps of Images from the ImageNette dataset

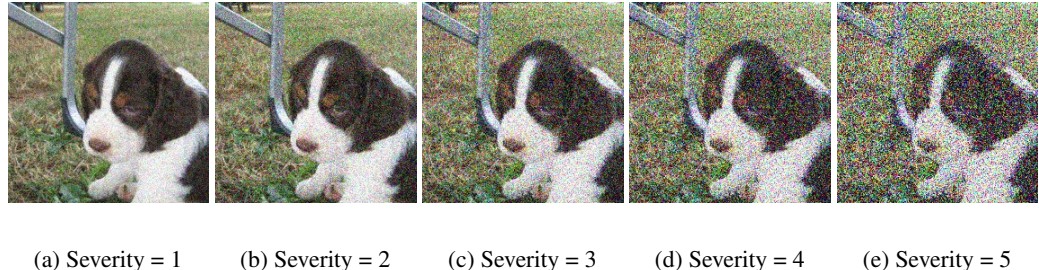

(a) Severity = 1     (b) Severity = 2     (c) Severity = 3     (d) Severity = 4     (e) Severity = 5

Figure 6:  Visualization of Images corrupted by Gaussian Noise (from ImageNet-C dataset)

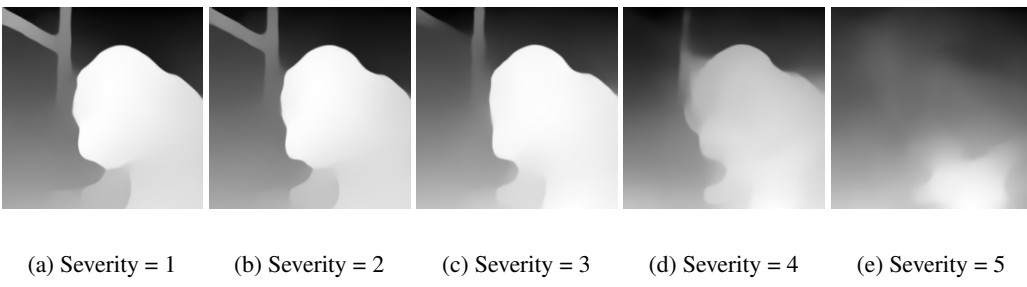

(a) Severity = 1     (b) Severity = 2     (c) Severity = 3     (d) Severity = 4     (e) Severity = 5

Figure 7:  Visualization of Depth Maps of Images corrupted by Gaussian Noise

Table 8: Different Augmentations on top of 3D Views.

| Method | Top-1 Acc. | IN-C | IN-3DCC |
|---|---|---|---|
| BYOL (Grill et al., 2020) | 85.27 | 84.13 | 83.68 |
| + 3D Views (Base SSL Aug) | 88.08 | 85.07 | 85.33 |
| + 3D Views (Minimal Aug) | 83.54 | 68.69 | 72.26 |

Table 9: Results on ImageNet-100 Noise Corruptions (IN-C). It can be clearly seen that the concatenation of the depth channel significantly improves the performance on noise based corruptions (by 8% in the case of Impulse noise). On the other hand, the introduction 3D Views hurts the performance on noise based corruptions.

| Method | IN-C | Gaussian Noise | Shot Noise | Impulse Noise | Speckle Noise |
|---|---|---|---|---|---|
| BYOL (Grill et al., 2020) | 47.15 | 37.08 | 36.00 | 28.31 | 45.36 |
| + Depth (p = 0.3) | 50.17 | 41.79 | 40.37 | 36.98 | 50.30 |
| + 3D Views | 48.15 | 34.25 | 33.25 | 27.04 | 43.46 |
| SimSiam (Chen & He, 2021) | 44.39 | 36.51 | 34.48 | 30.80 | 43.00 |
| + Depth (p = 0.2) | 48.30 | 41.36 | 39.98 | 36.99 | 49.29 |
| + 3D Views | 45.78 | 34.85 | 33.66 | 28.86 | 42.61 |

# E   ADDITIONAL RESULTS

**What happens when the base SSL augmentations are not applied on 3D Views?** Table 8 analyzes the role of augmentations applied on top of the synthesized 3D Views. "Base SSL Aug" refers to applying the same augmentations as the base SSL method, whereas "Minimal Aug" means that only Random Resized Crop and Horizontal Flip are used as augmentations. With 3D Views, even without the sophisticated augmentations, the model's linear evaluation performance is close to baseline BYOL trained with heavy augmentations.

Table 9 and 10 summarize the results on Noise Based Corruptions and Blur Corruptions respectively. Table 11 and 12 reports the results on Weather based and Digital Corruptions respectively.

Table 13 and Table 14 report the performance of corruptions in ImageNet-3DCC dataset.

Table 10: Results on ImageNet-100 Blur Corruptions (IN-C). Both the depth channel and 3D Views method improve the accuracy on blur based corruptions. The introduction of the 3D Views helps the model capture the 3D structure more easily and thus is highly robust to blur based corruptions.

| Method | IN-C | Defocus Blur | Glass Blur | Motion Blur | Zoom Blur | Gaussian Blur |
|---|---|---|---|---|---|---|
| BYOL (Grill et al., 2020) | 47.15 | 40.77 | 33.37 | 37.03 | 37.76 | 46.30 |
| + Depth (p = 0.3) | 50.17 | 40.21 | 36.89 | 38.50 | 41.55 | 46.16 |
| + 3D Views | 48.15 | 45.21 | 37.32 | 39.98 | 42.13 | 50.70 |
| SimSiam (Chen & He, 2021) | 44.39 | 36.84 | 30.92 | 34.72 | 35.32 | 42.76 |
| + Depth (p = 0.2) | 48.30 | 37.34 | 34.94 | 37.64 | 39.17 | 42.92 |
| + 3D Views | 45.78 | 40.58 | 35.21 | 39.19 | 41.40 | 45.72 |

Table 11: Results on ImageNet-100 Weather Corruptions (IN-C). The proposed method with the incorporation of depth channel results in a large increase on the performance of weather-corrupted images.

| Method | IN-C | Snow | Frost | Fog | Brightness |
|---|---|---|---|---|---|
| BYOL (Grill et al., 2020) | 47.15 | 35.93 | 41.79 | 46.84 | 73.71 |
| + Depth (p = 0.3) | 50.17 | 40.15 | 46.46 | 46.48 | 74.42 |
| + 3D Views | 48.15 | 38.43 | 42.48 | 45.99 | 73.72 |
| SimSiam (Chen & He, 2021) | 44.39 | 32.78 | 38.62 | 40.10 | 69.48 |
| + Depth (p = 0.2) | 48.30 | 38.84 | 44.11 | 45.81 | 70.78 |
| + 3D Views | 45.78 | 35.20 | 38.86 | 41.45 | 69.3 |

Table 12: Results on ImageNet-100 Digital Corruptions (IN-C). Combining the depth channel with the input improves the performance of all kinds of digital corruptions whereas we observe that 3D Views improves the accuracy on some corruptions and the performance degrades with some corruptions.

| Method | IN-C | Elastic | Contrast | Pixelate | Saturate | Spatter | JPEG |
|---|---|---|---|---|---|---|---|
| BYOL (Grill et al., 2020) | 47.15 | 53.32 | 50.57 | 65.94 | 71.92 | 51.02 | 63.22 |
| + Depth (p = 0.3) | 50.17 | 58.50 | 51.62 | 69.10 | 72.55 | 54.98 | 66.26 |
| + 3D Views | 48.15 | 58.74 | 50.32 | 66.73 | 69.79 | 51.26 | 63.97 |
| SimSiam (Chen & He, 2021) | 44.39 | 50.32 | 49.28 | 60.91 | 69.44 | 47.25 | 57.94 |
| + Depth (p = 0.2) | 48.30 | 55.37 | 49.95 | 66.08 | 69.54 | 53.33 | 64.14 |
| + 3D Views | 45.78 | 54.65 | 47.69 | 62.50 | 68.17 | 47.38 | 62.48 |

Table 13: Results on ImageNet-100 3D Corruptions (Subset of ImageNet-3DCC). Both the proposed methods improve upon the base SSL method in terms of the 3D Corruptions with the 3D Views being the best of the three.

| Method | IN-3DCC | Far Focus | Flash | Low Light | Near Focus | XY-Motion Blur | Z Motion Blur |
|---|---|---|---|---|---|---|---|
| BYOL (Grill et al., 2020) | 53.69 | 59.09 | 47.85 | 53.98 | 64.84 | 31.12 | 36.22 |
| + Depth (p = 0.3) | 55.55 | 60.42 | 50.24 | 57.37 | 65.18 | 34.28 | 42.04 |
| + 3D Views | 54.88 | 61.39 | 49.36 | 53.98 | 66.75 | 34.73 | 41.82 |
| SimSiam (Chen & He, 2021) | 50.44 | 55.31 | 44.82 | 48.51 | 61.67 | 28.93 | 34.34 |
| + Depth (p = 0.2) | 52.93 | 58.78 | 47.16 | 52.76 | 62.61 | 32.62 | 39.93 |
| + 3D Views | 52.17 | 57.24 | 45.94 | 48.88 | 63.27 | 34.10 | 42.29 |

Table 14: Results on ImageNet-100 3D Corruptions (Subset of IN-3DCC). Depth Channel improves upon the performance of non-3D corruptions like Iso-Noise and Color Quant.

| Method | IN-3DCC | Fog3D | Iso-Noise | Color Quant | Bit Error |
|---|---|---|---|---|---|
| BYOL (Grill et al., 2020) | 53.69 | 51.68 | 33.36 | 66.15 | 51.78 |
| + Depth (p = 0.3) | 55.55 | 50.64 | 39.15 | 67.44 | 52.09 |
| + 3D Views | 54.88 | 51.55 | 29.82 | 65.64 | 52.30 |
| SimSiam (Chen & He, 2021) | 50.44 | 48.26 | 32.56 | 62.42 | 48.69 |
| + Depth (p = 0.2) | 52.93 | 48.24 | 39.53 | 64.46 | 49.30 |
| + 3D Views | 52.17 | 48.83 | 30.87 | 63.13 | 48.72 |

