# OpenReview forum: "Leveraging the Third Dimension in Contrastive Learning"
_ICLR.cc/2023/Conference — Submitted to ICLR 2023_

### Official Review · Reviewer_nFbm · 2022-10-14

**Confidence:** 2
**Correctness:** 3
**Technical Novelty And Significance:** 3
**Empirical Novelty And Significance:** 3
**Recommendation:** 6

**Clarity, Quality, Novelty And Reproducibility:**

Overall, this paper is well-written even though some critical technical details are not discussed. The idea is interesting and useful.

**Strength And Weaknesses:**

**Strengths:**

(1). The idea is novel and interesting. I feel this is a very natural extension from the regular 2D image contrastive learning. This technique seems also very general and can be used in different CL backbones.

(2). The paper is well-written and easy to follow. I can understand the main idea by just reading the abstract.

(3). Experiments are strong. The improvements are not marginal.


**Weaknesses:**

(1). My main concern is that the monocular depth estimation usually needs a lot of human annotations (in pixels). In this case, when we really want to use the technique proposed in this paper, we will have to put lots of effort into annotation. More importantly, as we know that the effectiveness of a pretrained monocular depth estimation model may hevalily depend on the parameters/setting of the camera, we can not say whether the depth estimation is accurate for different datasets.

(2). Details should be further clarified. It is unclear how the model performs the estimation and how it generates the new views via modifying the camera position. These are important technical details, but they are missing.

(3). Since the idea is rather straightforward, I think the authors should consider releasing their code (maybe on some anonymous URL).

**Summary Of The Paper:**

This paper proposes to leverage the 3D information of natural images in contrastive learning. Specifically, the authors use the depth signal to generate multiple views from different camera positions, so that they can produce the 3D augmentation for contrastive learning. The idea is simple and natural, and validated on different popular baselines such as BYOL and SimSiam. Experiments on various styles of Imagenet demonstrate the effectiveness of the proposed methods.

**Summary Of The Review:**

I would like to vote for a "borderline accept" due to the above weaknesses I mentioned before. I would consider increasing my score if they are well solved.

---

> ### Author Response · Authors · 2022-11-12
> **Response to Reviewer nFbm**
>
> We thank the reviewer for the detailed feedback and encouraging comments on our paper.
>
> > My main concern is that the monocular depth estimation usually needs a lot of human annotations (in pixels). In this case, when we really want to use the technique proposed in this paper, we will have to put lots of effort into annotation. More importantly, as we know that the effectiveness of a pretrained monocular depth estimation model may hevalily depend on the parameters/setting of the camera, we can not say whether the depth estimation is accurate for different datasets.
>
> The pre-trained depth models are trained on datasets collected with RGB+D cameras and do not require any human annotations. Also, we would like to highlight that the current depth estimation models are highly generalizable and can be used without any fine-tuning. DPT reports remarkable performance in zero-shot cross dataset transfer. DPT models are not trained with the ImageNet dataset.
>
> > Details should be further clarified. It is unclear how the model performs the estimation and how it generates the new views via modifying the camera position. These are important technical details, but they are missing.
>
> We placed these details in Appendix B due to lack of space, but should have provided the reader of the main article with a clear pointer to the appendix.
>
> > Since the idea is rather straightforward, I think the authors should consider releasing their code (maybe on some anonymous URL).
>
> We have now released the code.
> The code is available at https://anonymous.4open.science/r/iclr2970-9A8F/

---

### Official Review · Reviewer_9hga · 2022-10-25

**Confidence:** 4
**Correctness:** 4
**Technical Novelty And Significance:** 2
**Empirical Novelty And Significance:** 2
**Recommendation:** 5

**Clarity, Quality, Novelty And Reproducibility:**

* The paper is well written and easy to follow.
* Experiments are well presented with sufficient ablation studies.

**Strength And Weaknesses:**

[Strength]

* Using RGB-to-Depth to get 3D presentation of image and then synthesizing a new view from 3D space is a novel idea in SSL.
* The proposed method improves the classification accuracy significantly on the benchmarked dataset.
* The idea is very simple and straightforward but very effective.

[Weakness]

* The novelty of this work is more or less incremental in nature. The authors combine two techniques from two deep learning domain. That is saying, this work itself dose not contribute any novel algorithm, except for the algorithm combining.

* Lack large-scale experiments as previous SSL works. ImageNet-100 is clearly too small for a serious study especially in SSL researches. Suggest to report results on ImageNet-1k and ImageNet-21k if possible.

* Possible information leak when using pre-trained RGB-to-Depth models. If the pre-trained models are trained on large-scale labeled dataset, using these pre-trained models is not fair then. This is because SSL aims to reduce the number of manual labels in deep learning. If this method requires a pre-trained model learned on an even larger-scale dataset, it simply makes no sense.


**Summary Of The Paper:**

The authors proposed a novel self-supervised learning method for 2D image classification. Instead of augmenting image in 2D space, the authors first use RGB-to-Depth to estimate the depth channel for the given input image. Then a new view of the image is synthesized from the RGB-D tensor. The synthetic views are then used in a contrastive learning framework. The authors validate their method on several small  scale datasets, including a subset of ImageNet.

**Summary Of The Review:**

The proposed method is motivated from a simple but interesting idea which mimics binocular vision system in self-supervised learning. The method shows promising improvement on small-scale image datasets but lacks large-scale experiments.

---

> ### Author Response · Authors · 2022-11-12
> **Response to Reviewer 9hga**
>
> We thank the reviewer for the comments and the interesting questions.
>
> > The novelty of this work is more or less incremental in nature. The authors combine two techniques from two deep learning domain. That is saying, this work itself dose not contribute any novel algorithm, except for the algorithm combining.
>
> We address the novelty in the general response.
>
> > Possible information leak when using pre-trained RGB-to-Depth models. If the pre-trained models are trained on large-scale labeled dataset, using these pre-trained models is not fair then. This is because SSL aims to reduce the number of manual labels in deep learning. If this method requires a pre-trained model learned on an even larger-scale dataset, it simply makes no sense.
>
> We're not certain we understand what the reviewer means by 'labels'. Neither DPT nor AdaMPI use class labels, of course. DPT does perform a supervised regression on depth values, but obtaining RGB+D data is very inexpensive. (Higher end phone cameras give an RGB+D input.) Also, whatever cost there is to train DPT is amortized over the fact that its predictions can be used for many tasks, among them our SSL task.
>
> We want to highlight these points from the introduction.
> “Depth is considered as the end product of vision, not a signal that can be exploited to improve visual information processing. Given the bias in favor of end-to-end models, researchers might suppose that if depth were a useful signal, an end-to-end computer vision system would infer depth.”

---

> > ### Comment · Reviewer_9hga · 2022-11-29
> > **Thank you for the author feedback and updated ImageNet-1k results**
> >
> > Thanks for the update! I still have concerns that cannot be simply addressed in this short time window:
> > 1) The novelty concern still remains. The idea is very interesting but a bit weak due to combination of two existing algorithms.
> > 2) I really appreciate the ImageNet-1k results. The improvement in "Linear" column is not significant. It does seem to be effective on ImageNet variant datasets though.

---

### Official Review · Reviewer_N68U · 2022-10-26

**Confidence:** 4
**Correctness:** 3
**Technical Novelty And Significance:** 2
**Empirical Novelty And Significance:** 2
**Recommendation:** 5

**Clarity, Quality, Novelty And Reproducibility:**

Clarity: The paper is easy to understand and clearly written
Quality and Novelty: Technical novelty is limited and empirical performance in some cases is good.
Reproducibility: I would suggest the authors to release the code during the review process.

**Strength And Weaknesses:**

Strengths:
- The paper is easy to follow and clearly written.
- The idea of incorporating the depth maps is interesting and to the best of my knowledge, it is the first attempt to use depth maps in the SSL framework.
- The proposed augmentation method and depth appended yields consistently better results on ImageNet100 and ImageNette dataset.

Weaknesses:
- The technical novelty in the proposed method is quite limited. The main contribution of the authors is to integrate existing approaches such as AdaMPI (Han et al. 2022) and DPT (Rangftl et al. 2021) in existing SSL procedures.
- As it’s an empirical paper, one would expect large scale experiments to test the proposed idea. The SSL experiments are performed on the subset of ImageNet dataset with 10 and 100 classes. To validate the idea, full Imagenet experiments will be needed. The main advantage of SSL methods is to learn on a large dataset of unlabeled images.
- The experiments are all performed on downstream tasks of classification. SSL literature also focuses on downstream tasks such as object detection and segmentation.
- In Table 2, adding 3D views results in worse performance. Can the authors explain their understanding why 3D views have negative effects on the SSL procedure? Why do some methods perform better at KNN based evaluation and some at Linear evaluation?
- In case of training with 3D views, are the novel views generated like depth maps beforehand or during training? Is there any computational overhead with the proposed method?



**Summary Of The Paper:**

The paper proposes to incorporate depth in the commonly used contrastive learning based self supervised learning frameworks. The depth map is generated using a state of the art RGB to depth predictor, namely DPT (Rangftl et al. 2021). The depth is simply appended to the input RGB image for the purpose of performing SSL procedure. Another contribution by the authors is in the use of AdaMPI (Han et al. 2022) to generate multiple views as an additional augmentation method. The overall  SSL procedure remains the same as the baseline method.


**Summary Of The Review:**

Since the technical novelty is limited and the main contribution is to combine two papers in terms of augmentation in existing SSL procedure, I think the paper is not good enough for publication in the conference. For an empirical contribution, I would expect authors to validate their idea with further analysis on large scale datasets.

Post-rebuttal:
Thanks to authors for the response to my comments. After reading the authors response and other reviewers concerns, I decide to stick to the original rating.

---

> ### Author Response · Authors · 2022-11-12
> **Response to Reviewer N68U**
>
> We thank the reviewer for their effort in providing a detailed review of our work.
>
> > The technical novelty in the proposed method is quite limited. The main contribution of the authors is to integrate existing approaches such as AdaMPI (Han et al. 2022) and DPT (Rangftl et al. 2021) in existing SSL procedures.
>
> We address the novelty in the general response.
>
> > In Table 2, adding 3D views results in worse performance. Can the authors explain their understanding why 3D views have negative effects on the SSL procedure?
>
> We thank the reviewer for this insightful question. We would like to emphasize that the quality of generated 3D Views plays a huge role in the performance. The ablation analyzing the quality of views on the performance provides evidence of the same (Table 7). We expect that as the quality of the generated images improves, the performance of SSL methods with 3D Views will improve as well.
>
> Also, we would like to highlight that we are not optimizing for the best hyperparameters in the view generation, unlike the base SSL method which has a carefully fine-tuned set of augmentations.
>
> > Why do some methods perform better at KNN based evaluation and some at Linear evaluation?
>
> The trend is usually consistent between kNN based evaluation and Linear evaluation with a few exceptions. Some of these trends are observed in other SSL works as well. For example, we refer the reviewer to Table 1 in [1]
>
> > In case of training with 3D views, are the novel views generated like depth maps beforehand or during training? Is there any computational overhead with the proposed method?
>
> Yes, the 3D Views are generated before training.
> There is no computational overhead with 3D views during training. The only computational overhead is due to the offline generation of 3D Views.
>
> > Reproducibility: I would suggest the authors to release the code during the review process.
>
> We have now released the code. The code is available at https://anonymous.4open.science/r/iclr2970-9A8F/
>
> [1] Adversarial Masking for Self-Supervised Learning, ICML 2022

---

### Official Review · Reviewer_TCcc · 2022-10-27

**Confidence:** 5
**Correctness:** 2
**Technical Novelty And Significance:** 2
**Empirical Novelty And Significance:** 2
**Recommendation:** 3

**Clarity, Quality, Novelty And Reproducibility:**

The paper is clearly written.
The paper combines the preexisting well performing methods.
Its code is not shared, therefore, it is not reproducible.

**Strength And Weaknesses:**

o	Strengths
	The paper is well written.
	Experimental results seem promising as the results in both 2 methods outperform the baseline results.
	Ablation experiments are well conducted although some experiments are missing.
o	Weaknesses
	There is no computational complexity information about adding the depth information in both methods. For the first method, concatenating a depth channel to the input, depth maps are pre-extracted before the training and used in augmentations, only in Random Crop and Horizontal flip transforms. In other types of transforms, depth map is assumed to be the same and as stated in the paper, it has a stronger correspondence.
Furthermore, only one of the two images is augmented (the other is the original image) rather than both augmented images in the originally proposed methods, SimSiam [1], BYOL [2] and SwAV [3]. Whether there is a specific reason to this choice is not described.
	Experiments are detailed except for the SwAV experiments in Table 1, ImagiNette, (+3D views are not presented.) and Table 2, ImageNet-100D, (there is no SwAV experiment.)
	This idea lacks of novelty. This paper can be considered as well performing components for each task, which are Depth Prediction Transformer, SwAV and AdaMPI but no new model is proposed.
	This is a nice application/engineering paper but ICLR is not the right venue for this paper.
	At least in ImageNette dataset, computing the depth map for each corresponding augmentation and training the methods as originally proposed should be presented. Also, SwAV experiments in Table 1 and 2 should be completed.

[1] Xinlei Chen and Kaiming He. Exploring simple siamese representation learning. In Proceedings of the IEEE/CVF Conference on Computer Vision and Pattern Recognition, pp. 15750–15758, 2021.
[2] Jean-Bastien Grill, Florian Strub, Florent Altche, Corentin Tallec, Pierre Richemond, Elena ´ Buchatskaya, Carl Doersch, Bernardo Avila Pires, Zhaohan Guo, Mohammad Gheshlaghi Azar, et al. Bootstrap your own latent-a new approach to self-supervised learning. Advances in neural information processing systems, 33:21271–21284, 2020.
[3] Mathilde Caron, Ishan Misra, Julien Mairal, Priya Goyal, Piotr Bojanowski, and Armand Joulin. Unsupervised learning of visual features by contrasting cluster assignments. Advances in Neural Information Processing Systems, 33:9912–9924, 2020.


**Summary Of The Paper:**

This paper proposes to integrate depth signal into self-supervised learning methods as an additional signal in 2 ways. The first method simply concatenates the depth channel onto the RGB image and trains the SSL methods with RGBD input. The second method uses the single-view view synthesis methods to generate new views and use them as augmentations. Experiments show significant improvements over the baseline methods especially in corruption datasets, ImageNet-C and ImageNet-3DCC.

**Summary Of The Review:**

As stated above, although this is a nice application/engineering paper, ICLR is not the right venue for this paper. It’s more suitable for a computer vision conference.

---

> ### Author Response · Authors · 2022-11-12
> **Response to Reviewer TCcc**
>
> We thank the reviewer for their effort in providing a detailed review of our work.
>
> > There is no computational complexity information about adding the depth information in both methods.
>
> We provide the compute time for generating the depth maps and 3D views. Kindly note that both of these are done offline (i.e., only once before training).
>
> *Depth Map Generation on ImageNette dataset with DPT Large (RTX 8000)*
>
> Overall time for 9469 images:  251.91 seconds \
> Average of each image:  0.0215 seconds
>
> 3D Views (with AdaMPI)\
> From the AdaMPI paper: Our method with 64 planes takes 0.072s, including 0.004s for depth adjustment and 0.068s for color prediction, to generate the MPI for a 256 × 384 image on a Nvidia Tesla V100 GPU.
>
> It should be noted that both the depth map generation and 3D Views can be easily parallelized.
>
> We also discuss the computational overhead of training with the proposed methods.
> Since the depth information is computed offline, the computational overhead of training any SSL with depth is negligible. Only the first layer of the backbone is modified to allow for a 4 dimensional input (RGB+D).
>
> The computational cost of training with 3D Views is similar to the training of the base SSL method as the 3D Views are generated offline.
>
> > For the first method, concatenating a depth channel to the input, depth maps are pre-extracted before the training and used in augmentations, only in Random Crop and Horizontal flip transforms. In other types of transforms, depth map is assumed to be the same and as stated in the paper, it has a stronger correspondence.
>
> Apart from Random Crop and Horizontal Flip transforms, the other types of augmentations which are used include Color Jitter, GrayScale, Gaussian Blur. In an ideal scenario, we do not expect the depth information to change when such augmentations are added.
>
> > Furthermore, only one of the two images is augmented (the other is the original image) rather than both augmented images in the originally proposed methods, SimSiam [1], BYOL [2] and SwAV [3]. Whether there is a specific reason to this choice is not described
>
> This is incorrect. We augment both the views with the depth maps. We apologize for the confusion.
>
> > Experiments are detailed except for the SwAV experiments in Table 1, ImagiNette, (+3D views are not presented.) and Table 2, ImageNet-100D, (there is no SwAV experiment.)
>
> We sincerely apologize for the confusion here. SwAV introduces the concept of multi-crop in its work where 2 high resolution views and 6 low resolution views are sampled from a particular image. We felt that the extension of the multi-crop was not straightforward in the case of 3D Views and that was the primary reason for not adding SwAV+3D Views.
>
> > This idea lacks of novelty. This paper can be considered as well performing components for each task, which are Depth Prediction Transformer, SwAV and AdaMPI but no new model is proposed.
>
> We address the novelty aspect in the general response.
>
> > As stated above, although this is a nice application/engineering paper,  ICLR is not the right venue for this paper. It’s more suitable for a computer vision conference.
>
> We politely disagree with the reviewer's comment. The paper is fundamentally about visual representation learning (the LR in ICLR). Our methods are designed with the goal of improving the representations and robustness of SSL methods.
>
> In previous editions too, ICLR has accepted works on improving visual representation learning [1].
>
> [1] Xiao, Tete, et al. "What should not be contrastive in contrastive learning." ICLR 2021.

---

### Author Response · Authors · 2022-11-12
**General Response: Novelty of our work**

The reviewers express concern about the lack of novelty in the work. While we acknowledge that our core idea is quite simple and elegant, its success was non-obvious and it offers a conceptual shift away from the predominant research strategy in the field.  We elaborate.

First, consider the premise of our experiments. Based on insights from cognitive neuroscience, we hypothesized that including estimated depth from RGB would improve representation learning. Prior to our experiments, we were skeptical that we would see benefits, as were our colleagues. Their typical response was that if depth were useful for representation learning, it would be extracted in end-to-end training. The reviewers should ask themselves whether the result -- which may seem obvious in retrospect -- was what they would have predicted. The robust improvements, across multiple SSL algorithms, are surprising and suggestive.

* Our work clearly shows that cognitive inductive bias can be helpful in SSL methods.

* Motivated by binocular vision, we propose to use existing View Synthesis and use these 3D Views as augmentations in SSL. We are not aware of any prior work that uses these synthetically generated views to improve representations. Again, we want to highlight the conceptual novelty and not the use of specific method like AdaMPI.

* We are not aware of any prior work that has explored the role and importance of 3D knowledge into the SSL framework.

* Most SSL methods require prior information on augmentations. For instance, random crops have a negative impact in object-centric representation learning [1]. On the other hand, NeRF and other View Synthesis based models have shown remarkable results in visualizing the scenes from different angles. With the increasing growth of generative models, it's possible that the future work is able to get rid of these handcrafted augmentations used in prior SSL methods. Our work can be seen as a step in this direction.

[1] Baldassarre, F., & Azizpour, H. (2022). Towards Self-Supervised Learning of Global and Object-Centric Representations. arXiv preprint arXiv:2203.05997.

---

### Author Response · Authors · 2022-11-17
**General Response: Large Scale Experiments**

Reviewers N68U and 9hga mentioned the importance of evaluating the ideas on large-scale datasets like ImageNet. We agree with the reviewers that the main advantage of SSL methods is on large datasets of unlabeled images. Despite the limited time and our limited resources, we have run initial experiments on a larger dataset, ImageNet-1k.

We observe that the observations scale up to ImageNet-1k. We observe significant accuracy boosts in classification of corrupted images: 1.5% for  ImageNet-C and 1.8% for ImageNet-3DCC. This further strengthens the argument about the role of depth channel and 3D Views in SSL methods.

The SimSiam model and SimSiam+Depth (p=0.2) is trained for 800 epochs.

| Method  | Linear Eval | IN-C | IN-3DCC |
| ----------- | ----------- | ----------- | ----------- |
| SimSiam      | 71.7  | 36.45 | 43.32 |
| SimSiam + Depth (p=0.2)     | 71.3  | 38.23 | 45.11 |

But due to limited time, we are reporting the SimSiam + 3D Views experiments on 100 epochs.

| Method  | Linear Eval | IN-C | IN-3DCC |
| ----------- | ----------- | ----------- | ----------- |
| SimSiam      | 68.10  | 32.99 | 38.94 |
| SimSiam + 3D Views   | 68.08  | 34.43 | 40.71 |

---

### Decision · Program_Chairs · 2023-01-20

**Decision:**

Reject

**Justification For Why Not Higher Score:**

The current recommendation is based on the weaknesses summarized above: in particular, limited technical novelty and lack of systematic large-scale evaluation.

**Justification For Why Not Lower Score:**

N/A.

**Metareview: Summary, Strengths And Weaknesses:**

Summary: This paper improves existing contrastive-based self-supervised learning (SSL) methods by integrating additional depth signals. Two strategies are investigated: (1) A prior RGB-to-depth prediction model DPT (Rangftl et al. 2021) is used to generate depth maps, and the depth channel is then concatenated onto the RGB channels; (2) images from new viewpoints are synthesized via a prior single-view synthesis method AdaMPI (Han et al. 2022), and then used as augmentations. Once the augmentation is generated, a typical SSL procedure follows. Experiments show improvements over baselines without using depth signals in small-scale 2D image classification experiments (e.g., subsets of ImageNet). During the discussion phase, some large-scale experimental results on ImageNet-1k are provided.

Strengths and Weaknesses: The reviewers generally acknowledge that it is an interesting direction to explore how to leverage 3D information or depth signals to improve self-supervised learning, but they have concerns about the novelty of specific methods in the paper. In particular, the techniques used in this paper for obtaining depth signals directly come from prior work on RGB-to-depth prediction DPT (Rangftl et al. 2021) and novel-view synthesis AdaMPI (Han et al. 2022), leading to incremental contribution of algorithm combination.

In addition, the experiments conducted in the original paper are small-scale. While the reviewers appreciate the additional large-scale experiments provided in the discussion phase, they feel that those experiments are preliminary. Also, the improvement in the linear probe setting seems not significant. More systematic large-scale evaluation is needed to validate the proposed approach.

Since additional pre-trained models are used to obtain depth signals, the extra needed supervision and computational cost from these models are not clearly discussed in the original paper. The authors clarify these in the discussion phase. Following this, it is important to clearly state the supervision and data needed to train the monocular depth estimation models, and why they do not break the standard SSL setting. And also, a comprehensive study is needed regarding how precise the obtained depth information should be or how the SSL performance would change with respect to the accuracy of depth information. There is some ablation in the paper, but a more in-depth study would be helpful to understand the behavior of the proposed approach.

The experiments are conducted on 2D image classification. The reviewers point out that SSL literature also focuses on downstream tasks such as object detection and segmentation. Following this comment, it would be interesting to investigate downstream tasks that rely on depth information, which may further validate the effectiveness of the proposed approach.

Therefore, the paper is not ready for this ICLR. I encourage the authors to continue this line of work for future submission.